# A Case-Control Analysis of the Impact of Venous Thromboembolic Disease on Quality of Life of Patients with Cancer: Quality of Life in Cancer (Qca) Study

**DOI:** 10.3390/cancers12010075

**Published:** 2019-12-26

**Authors:** Lucia Marin-Barrera, Andres J. Muñoz-Martin, Eduardo Rios-Herranz, Ignacio Garcia-Escobar, Carmen Beato, Carme Font, Estefania Oncala-Sibajas, Alfonso Revuelta-Rodriguez, Maria Carmen Areses, Victor Rivas-Jimenez, Maria Angeles Moreno-Santos, Aitor Ballaz-Quincoces, Juan-Bosco Lopez-Saez, Iria Gallego, Teresa Elias-Hernandez, Maria Isabel Asensio-Cruz, Leyre Chasco-Eguilaz, Gonzalo Garcia-Gonzalez, Purificacion Estevez-Garcia, Remedios Otero, Jorge Lima-Alvarez, Luis Jara-Palomares

**Affiliations:** 1Respiratory Department, Medical Surgical Unit of Respiratory Diseases, Hospital Virgen del Rocio, CIBERES, 41013 Sevilla, Spain; luciamarinb@gmail.com (L.M.-B.); teresaelias@telefonica.net (T.E.-H.); miacruz06@hotmail.com (M.I.A.-C.); rotero@separ.es (R.O.); 2Medical Oncology Department, Hospital General Universitario Gregorio Marañón, 28009 Madrid, Spain; andresmunmar@hotmail.com (A.J.M.-M.); iriagallegog@gmail.com (I.G.); tripleg87@yahoo.es (G.G.-G.); 3Hematology and Hemotherapy Department, Hospital Universitario Virgen de Valme, 41701 Sevilla, Spain; eriosh@gmail.com; 4Medical Oncology Department, HGU de Ciudad Real, 13003 Ciudad Real, Spain; naxto@hotmail.com; 5Medical Oncology Department, Hospital Universitario Virgen Macarena, 41009 Sevilla, Spain; cbeatoz@hotmail.com; 6Medical Oncology Department, DIBAPS/Translational Genomics and Targeted Therapeutics in Solid Tumors, Hospital Clínic, 08036 Barcelona, Spain; carme.fontpuig@gmail.com; 7Emergency Department, Hospital Virgen Macarena, 41009 Sevilla, Spain; estefania.oncala@gmail.com; 8Medical Oncology Department, Hospital Universitario Central de Asturias, 33611 Oviedo, Spain; revuelta.al@gmail.com; 9Medical Oncology Department, Hospital de Orense, 32616 Orense, Spain; karmeleareses@hotmail.com; 10Medical Oncology Department, Hospital de Jerez de la Frontera, 11407 Cádiz, Spain; victorjjrivas@hotmail.com; 11Medical Oncology Department, H. U. Puerto Real, 11510 Cádiz, Spain; angeles.moreno@uca.es; 12Respiratory Department, Hospital de Galdakao-Usansolo, 48960 Bizkaia, Spain; aitor.ballazquincoces@osakidetza.net (A.B.-Q.); leyrech10@gmail.com (L.C.-E.); 13Internal Medicine Unit. H. U. Puerto Real, 11510 Cádiz, Spain; juanbosco.lopez@gm.uca.es; 14Medical Oncology Department, Hospital Virgen del Rocio, 41013 Sevilla, Spain; puriestevez@gmail.com; 15Respiratory Department, Hospital Universitario Virgen de Valme, 41701 Sevilla, Spain; jorgelial@hotmail.com

**Keywords:** neoplasm, quality of life, venous thromboembolism, pulmonary embolism, venous thrombosis

## Abstract

Although there is published research on the impact of venous thromboembolism (VTE) on quality of life (QoL), this issue has not been thoroughly investigated in patients with cancer—particularly using specific questionnaires. We aimed to examine the impact of acute symptomatic VTE on QoL of patients with malignancies. This was a multicenter, prospective, case-control study conducted in patients with cancer either with (cases) or without (controls) acute symptomatic VTE. Participants completed the EORTC QLQ-C30, EQ-5D-3L, PEmb-QoL, and VEINES-QOL/Sym questionnaires. Statistically significant and clinically relevant differences in terms of global health status were examined. Between 2015 and 2018, we enrolled 425 patients (128 cases and 297 controls; mean age: 60.2 ± 18.4 years). The most common malignancies were gastrointestinal (23.5%) and lung (19.8%) tumors. We found minimally important differences in global health status on the EQ-5D-3L (cases versus controls: 0.55 versus 0.77; mean difference: −0.22) and EORTC QLQ-C30 (47.7 versus 58.4; mean difference: −10.3) questionnaires. There were minimally important differences on the PEmb-QoL questionnaire (44.4 versus 23; mean difference: −21.4) and a significantly worse QoL on the VEINES-QOL/Sym questionnaire (42.7 versus 51.7; mean difference: −9). In conclusion, we showed that acute symptomatic VTE adversely affects the QoL of patients with malignancies.

## 1. Introduction

Venous thromboembolic disease (VTE) consists of pulmonary embolism (PE) and deep vein thrombosis (DVT). In the general population, PE ranks third among all causes of cardiovascular mortality—following myocardial infarction and stroke [1]. The burden of this condition is even higher in patients with malignancies [2]. Patients with tumors also present a markedly increased risk (up to a four-fold) of VTE than those without. Conversely, approximately 20% of patients with VTE have an active malignancy [3], which in turn portends a higher risk of complications (i.e., bleeding, recurrent VTE, and death) [4,5,6,7].

Health-related quality of life (QoL)—defined as the perception a patient has of the effects of a disease or its treatment on distinct domains, including physical, emotional, and social well-being [8]—has recently become a subject of intense investigation. Several generic questionnaires aimed at assessing QoL in everyday clinical practice have been developed. However, disease-specific questionnaires are generally more sensitive in the detection of subtle QoL changes in selected clinical populations. Different questionnaires are currently available for the specific assessment of QoL in patients with VTE. The Venous Insufficiency Epidemiological and Economic Study-Quality of Life/Symptoms (VEINES-QOL/Sym) [9,10] and the Pulmonary Embolism Quality of Life questionnaire (PEmb-QoL) [11,12] have been designed for patients with DVT and PE, respectively.

Although there is published research on the impact of DVT [13] and PE [14] on QoL, this issue has not been thoroughly investigated in patients with cancer [15] and, particularly, using specific questionnaires. We therefore designed the Quality of life in Cancer (QCa) study to examine the impact of acute symptomatic VTE on QoL of patients with malignancies.

## 2. Methods

### 2.1. Study Design

The QCa study is a multicenter, prospective, case-control investigation conducted in patients with malignancies. Recruitment was conducted between June 2015 and January 2018 in the outpatient facilities of 13 Spanish hospitals. The study investigators were in specialists in oncology, hematology, internal medicine, and respiratory medicine. Patients with cancer either with (cases) or without (controls) acute symptomatic VTE were included. Controls were deemed eligible at any time during the course of their disease and were required to complete the QoL questionnaires at inclusion. Cases were included in the study at the time of acute symptomatic VTE, and quality of life were filled 30 days after the index thrombotic event (30 ± 2 days). All questionnaires were completed at the same time point to avoid a source of potential confounding. All clinical data and response to questionnaires were separated from personal information and analysed in an anonymized fashion. The study protocol was reviewed and approved by the Research Ethics Committee of the Virgen del Rocio Hospital, Sevilla, Spain (code: 0191-N-14) and subsequently ratified by each participating center. All patients provided written informed consent before inclusion. The complete protocol has been previously described in detail [16].

The main research of the study was to investigate the impact of acute symptomatic VTE on QoL of patients with malignancies. With this aim, we compared the global health status of cases and controls using four QoL questionnaires. Secondary endpoints included the comparisons of different QoL dimensions and three subgroup analyses (PE versus DVT, men versus women, analysis according to tumor site). The following questionnaires were used: (1) European Organization for Research and Treatment of Cancer quality of life questionnaire (EORTC QLQ-C30) version 3.0 [17], which is designed to assess QoL in patients with cancer; (2) the generic EQ-5D-3L questionnaire [18]; (3) the PEmb-QoL questionnaire, which is specific for patients with PE [11], and (4) the VEINES-QOL/Sym questionnaire [9,10], which examines specific symptoms of DVT and focuses on limitations of daily living activities. 

The EQ-5D-3L questionnaire is a commonly used generic health questionnaire capable of comparing different diseases [18]. It allows the calculation of a utility index (UI), which comprises five different dimensions (mobility, self-care, daily activities, pain/discomfort, and anxiety/depression) [18] which are scored from 1 (no problems) to 5 (unable to/extreme problems). The responses are combined to obtain a 5-digit number (from 11, 111 to 55555) that summarizes the patient’s health status. This value is subsequently converted into the UI, which may range from 0.208 (worst possible health) to 1.000 (best possible health). The minimally important difference (MID) on the EQ-5D questionnaire for patients with malignancies is 0.06−0.08 UI [19].

The EORTC QLQ-C30 questionnaire, which has been specifically designed for patients with malignancies, comprises five functioning scales (physical, social, role, cognitive, and emotional functioning), three symptom scales (fatigue, pain, and nausea/vomiting), one global health scale, and six independent items (dyspnea, sleep disturbances, anorexia, constipation, diarrhea, and economic impact). High scores on the global health scale and functioning status reflect a better QoL, whereas high values in the symptoms scale indicate a reduction in QoL (owing to the presence of cancer-associated symptoms) [20,21,22]. The index score ranges from 0 (death) to 1 (full health), with higher scores (from 0.654 to 1) reflecting a better health status. The MID on the EORTC QLQ-C30 questionnaire varies from 6 to 15 points [23,24,25].

The PEmb-QoL questionnaire—a disease-specific questionnaire for patients with PE [11]—has six different dimensions (frequency of complaints, limitations in daily life activities, work-related problems, social limitations, intensity of complaints, and emotional complaints). Higher scores reflect a lower QoL and the minimally clinically important difference (MCID) is 15 points [26]. The VEINES-QOL/Sym questionnaire [9,10], which takes into account the symptoms, limitations, and daily life activities of the last four weeks, provides the VEINES-QOL score (which reflects QoL) and the VEINES/Sym score (which provides a quantitative measure of symptoms). Higher scores reflect a better QoL.

### 2.2. Sample Size Calculation

The planned case-control ratio was 1:2. To achieve a power of 80% to detect differences (in contrast to the null hypothesis) using a two independent-sample Student’s *t*-test with an alpha error of 0.05, under the assumptions of a mean in the control group of 55.75 units, a mean in the case group of 45.26 units, and a standard deviation in both groups of 26.05 units, a total of 98 cases and 196 controls were required (totaling 294 subjects). Under the hypothesis of a 5% dropout rate, we planned to recruit a minimum of 104 cases and 208 controls (totaling 312 subjects).

### 2.3. Statistical Analysis

For each questionnaire, the distributions of scores, means, medians, and the percentage of respondents with minimum/maximum scores (indicating flooring and ceiling effects) are presented. The criteria of acceptability for each questionnaire were as follows: (1) less than 10% missing data for summary scores, (2) even distribution of endorsement frequencies across response categories, and (3) less than 10% floor and ceiling effects for summary scores. We applied the relative efficiency (RE) statistic and calculated the effect size (Cohen’s d) to estimate the questionnaire performance (i.e., sensitivity to detect differences) in each study group. The RE statistic was defined as the ratio of F-statistics in the analysis of variance (ANOVA) tests of the differences in scores between known groups [27]. Higher F-statistic values indicate a greater efficiency or discrimination (versus the comparator). The effect size was calculated using the difference in mean scores divided by the pooled standard deviation. The threshold values for effect sizes were defined as small (0.2), moderate (0.5), and large (0.8). Figure 1 provides a violin plot. Violin plots allow plotting numerical data and can be regarded as a combination between box plots and kernel density plots. Kernel density estimation (KDE) is a non-parametric approach to estimate the probability density function of a random variable. Moreover, these plots show the actual distribution of the observed values (dots in cases and triangles in controls), ultimately allowing a representation of the scores used in the study. Boxplots depicts the median (black line inside the box) and the interquartile range (the edges of the box). The kernel density plot (grey line) represents the probability density function. Larger sections of the violin plot indicate a higher probability of observation at a given value, whereas thinner sections correspond to a lower probability. Statistical calculations were performed using R (version 3.5.3; March 11, 2019) and R Studio (version 1.1.456) equipped with the dplyr (version 0.8.3), effsize (version 0.7.4), PMCMR (version 4.3), catspec (version 0.97), and ggplot2 (version 3.1.0) packages. The full reproducible code is provided in the Appendix A.

### 2.4. Data Sharing Statement

For original data, please contact luisoneumo@hotmail.com. Individual participant data will not be shared.

## 3. Results

The study sample consisted of 425 patients (128 cases and 297 controls; mean age: 60.2 ± 18.4 years; 58% men) (Appendix A include recruitment per center). Of them, 77% were receiving anticancer treatment and 20% had a central venous catheter. The most common malignancies were gastrointestinal (23.5%) and lung (19.8%) tumors. Distant metastases were identified in 68% of the study patients, and the Eastern Cooperative Oncology Group (ECOG) performance status was 0–1 in 89% of the participants (Table 1). Adenocarcinoma was the most frequent histology type (58.8%). The distribution of VTE was as follows: DVT (54.7%), PE (25%), and DVT plus PE (20.3%). There were no significant differences between the time from cancer diagnosis and the inclusion in the study between cases (14.76 ± 21.34 months) and controls (15.01 ± 25.57 months). More than half of all cases (54.5%) and controls (58.1%) were included within 6 months from cancer diagnosis. All cases received weight-adjusted low molecular weight heparin at inclusion. Appendix A shows the marital status, education status, employment status, and gross income of the study participants.

### 3.1. Impact of Venous Thromboembolism on Quality of Life of Patients with Malignancies

The presence of thrombosis in patients with malignancies had a statistically significant and clinically relevant negative impact on the quality of life according to the results of four different questionnaires. Of the 425 patients included, 84% completed the EQ-5D questionnaire (n = 355), 83% the EORTC QLQ-C30 questionnaire (n = 354), 72% the Pemb-QOL questionnaire (n = 306), and 72% the VEINES-QOL/Sym questionnaire (n = 305). Cases and controls differed significantly in terms of EQ-5D-3L results, and such difference was larger than the MID (0.55 versus 0.77, respectively; mean difference: −0.22). Similar results were obtained on the EORTC QLQ-C30 questionnaire with regard to the global health status (47.7 versus 58.4, respectively; mean difference: −10.3), physical functioning (64.4 versus 80.6, respectively; mean difference: −16.2), role functioning (53.9 versus 74.7, respectively; mean difference: −20.8), and social functioning (62.2 versus 74.1, respectively; mean difference: −11.9). As far as symptoms are concerned, cases scored higher in terms of fatigue (48.4 versus 33.7, respectively; mean difference: +14.7), pain (36 versus 23, respectively; mean difference: +13). and dyspnea (23.5 versus 13, respectively; mean difference: +10.5; Table 2).

Cases and controls differed significantly in terms of PEmb-QoL results, and such difference was larger than the MID with regard to QoL (44.4 versus 23, respectively; mean difference: −21.4) and five of the six measured dimensions. Similarly, the VEINES-QOL/Sym questionnaire showed worse QoL in cases than in controls (42.7 versus 51.7, respectively; mean difference: −9), with the former group having more symptoms (43.8 versus 51.4, respectively; mean difference: −7.6). Figure 1 shows four sections of the violin plot indicating differences in the median between cases and controls (*p* < 0.05). There were also marked differences in kernel distribution. Specifically, data from controls surrounded the median values for the four questionnaires, whereas a wider distribution was observed for cases.

### 3.2. Subgroup Analysis

No significant differences in terms of EQ-5D-3L results were found between patients with PE versus DVT. However, the two groups differed significantly with regard to dyspnea as measured by the EORTC QLQ-C30 questionnaire, and such difference was larger than the MID (32.7 versus 15.9; mean difference: −16.8; Appendix A).

We subsequently performed an age- and sex-stratified analysis of the EORTC QLQ-C30 questionnaire in cases and controls. Regardless of sex, we found statistically significant differences larger than the MID in the great majority of the functioning scales (Appendix A). Similar differences were observed on the EQ-5D-3L questionnaire. As far as the PEmb-QoL questionnaire is concerned, the presence of PE was found to affect QoL (with the difference being larger than the MCID) in women but not in men (49.9 versus 20.1, respectively; mean difference: −29.8). Similar findings were observed with regard to all of the dimensions. Only work-related problems and the intensity of complaints were found to be affected in men. In contrast, the VEINES-QOL/Sym questionnaire showed statistically significant differences in both sexes.

When QoL was analyzed in relation to the cancer site, the functioning scales of the EORTC QLQ-C30 questionnaire showed statistically significant differences (larger than the MID) in several dimensions among patients with gynecologic and digestive tumors. Patients with gynecologic malignancies also showed differences in global health status, physical status, role, and social functioning. With regard to digestive tumors, significant differences were observed in physical status, role, emotional, and cognitive functioning (Appendix A). An analysis of symptoms on the EORTC QLQ-C30 questionnaire is provided in Appendix A. The EQ-5D-3L questionnaire yielded statistically significant differences (larger than the MID) for all tumor sites, the only exception being lung cancer (Appendix A). In the subgroup of patients with DVT, the VEINES-QOL/Sym questionnaire identified statistically significant differences in both QoL and symptoms for patients with digestive and pulmonary tumors (Appendix A). The results on the PEmb-QoL questionnaire were also found to differ according to the tumor site (Appendix A).

## 4. Discussion

The present case-control analysis demonstrates a significant detrimental impact of acute symptomatic VTE on QoL of patients with malignancies, with a worse global health status being detected by all of the four questionnaires used in our study. There were differences larger than the MID on the EQ-5D-3L and the EORTC QLQ-C30 questionnaires and greater than the MCID on the PEmb-QoL questionnaire. Moreover, a statistically significant worse QoL was identified on the VEINES-QOL/Sym questionnaire. Using the EORTC QLQ-C30 questionnaire, we similarly showed that acute symptomatic VTE is associated with lower physical functioning, role functioning, social functioning, as well as higher fatigue, pain, and dyspnea. A detrimental impact was also observed on five of the six dimensions of the PEmb-QoL questionnaire. Our findings have practical implications. Owing to the negative impact of thrombosis on the quality of life of patients with malignancies, healthcare personnel should be aware of this aspect and pay adequate attention to its detection. In addition, our data may pave the way to future research aimed at assessing the impact on quality of life of treatments and interventions implemented in patients with cancer-associated thrombosis.

According to the World Health Organization, QoL should be considered as a multidimensional concept [28]. Generic QoL questionnaires allow comparing of different diseases or processes, whereas disease-specific questionnaires can serve as more sensitive tools for analyzing the repercussions of a given disease on the patient. Recent years have witnessed a mounting interest in QoL as a healthcare outcome measure—with increasingly more studies focusing not only on the quantity but also on the quality of lives lived [29]. 

In a study conducted in 359 patients with DVT, Kahn et al. [13] investigated QoL by administering the 36-Item Short-Form Health Survey (SF-36) and the VEINES QOL/Sym questionnaires at baseline and at two different follow-up visits (at 1 and 4 months). The results of the VEINES QOL/Sym questionnaire demonstrated a slightly higher QoL in their sample compared with our study (50 versus 42.7, respectively). This difference may be explained by the fact that only 12.5% of their study patients had cancer [13]. Tavoly et al. [14] performed a cross-sectional study of QoL in 213 patients with PE—with their results showing a higher EQ-5D index score than that observed in our study (0.8 versus 0.55, respectively). Such difference may be attributed to the fact that only 7% of their study participants had a diagnosis of cancer. In addition, questionnaires were administered after 4 months and 10 years from the index PE event [14]. The prospective, observational, international PREFER study investigated the QoL of 1399 patients with PE using the EQ-5D-5L questionnaire‚—with a reported score of 0.71 ± 0.27 at the time of event [30]. Only 120 patients (8.6%) in this study had a diagnosis of cancer and disease-specific QoL questionnaires were not utilized. The QUAVITEC study was a prospective, longitudinal study of patients with cancer and VTE [31] who were administered three questionnaires on QoL (SF-36, EORTC QLQ-C30, and VEINES-QOL) at baseline and at two follow-up visits (3 and 6 months). Although a better QoL was reported at 6 months after the index VTE event, these results should be interpreted cautiously because only 49% of the study patients underwent the 6-month assessment. Notably, the global health status/QoL score on the EORTC QLQ-C30 questionnaire was similar to that observed in our current investigation (47 versus 47.7, respectively). A sub-analysis of the CATCH trial has been published in 2008 [32]. The study investigated the utility of LMWH tinzaparin in preventing recurrent VTE in patients with different types of cancer and the EQ-5D questionnaire was administered to the participants. The authors found that recurrent VTE had a significant impact on EQ-5D scores (decrease: −0.075). The strengths of this study included the large sample size (n = 883) and the repeated monthly measures of quality of life for 7 months. However, this research also had some inherent caveats, including the use of a generic—and not disease-specific—questionnaire and the comparison with a predefined patient profile (male, from Western Europe, without distant metastases, with symptomatic DVT as the qualifying event, and an ECOG performance status of 1 at baseline).

Data obtained from quality of life questionnaires provide clinically relevant and reproducible information with respect to different patient dimensions. These results may be also complemented with qualitative research work. In this regard, a previous study conducted semi-structured interviews in 14 patients with cancer-associated thrombosis [33]. The results indicated not only that VTE had a major adverse impact on their life but also that thrombosis was considered a distinct entity—rather than a part—of their cancer illness. The following three areas were found to be affected by VTE: (1) symptom burden; (2) impact in the context of their cancer journey; (3) impact on their activities of daily living.

Our study has several strengths. First, we specifically focused on the impact of acute symptomatic VTE on QoL in patients with cancer using different questionnaires. This approach allowed us to compare our findings with those obtained in other diseases (owing to the use of the generic EQ-5D-3L questionnaire), in other cancer-associated complications (owing to the use of the EORTC QLQ-C30 questionnaire), and in other clinical populations with VTE (owing to the use of the PEmb-QoL and VEINES-QOL questionnaires). Second, our case-control design allowed us obtaining direct comparisons of QoL, without resorting to historical cohorts characterized by different demographic and clinical characteristics. Third, the comparative analysis of QoL conducted in our study took into account either MID or MCID. We were therefore able to identify clinically relevant differences that were not merely statistically significant. 

However, our findings need to be interpreted in the context of some limitations. First, although the design of our work was a case-control study, the prospective and multicenter nature of our case-control study did not allow us obtaining a perfect matching for all variables that may potentially affect quality of life. For example, tumor location was not well balanced between cases and controls. In any case, thrombosis was found to have a negative impact on quality of life regardless of tumor location. In addition, more than half of all cases (54.5%) and controls (58.1%) were included within 6 months from cancer diagnosis. Also, the metastasis rate did not show significant intergroup differences. Moreover, there were other variables that could affect to QoL that were not included in our study (i.e., chemotherapy use, episodes of neutropenic fever, infections, recent hospitalizations, if patients received curative or palliative treatment or 12-month survival probability in both groups). Second, the number of participants was not sufficiently large to detect statistically significant differences in certain questionnaire dimensions or specific subgroups. This caveat despite that predetermined sample size was sufficiently large to demonstrate an impact of acute symptomatic VTE on QoL of patients with malignancies—a hypothesis that was ultimately confirmed. Third, we did not perform serial assessments of QoL in our sample. Although this methodological aspect may be essential in patients with VTE and no malignancies, its importance is less paramount in the oncology setting. Accordingly, the dismal prognosis of patients with metastatic cancer may lead to a bias similar to that occurring in the QUAVITEC study, which showed an improved QoL at a 6-month follow-up, albeit at the expenses of a limited number of responders (less than 50% compared with baseline) [31].

## 5. Conclusions

In summary, the results of our study demonstrate that acute symptomatic VTE has an adverse impact on the QoL of patients with malignancies. Clinicians who treat cancer-associated thrombosis should be aware that this event is not merely a complication but has a broader impact on the patient’s quality of life, ultimately requiring a comprehensive clinical management.

## Figures and Tables

**Figure 1 cancers-12-00075-f001:**
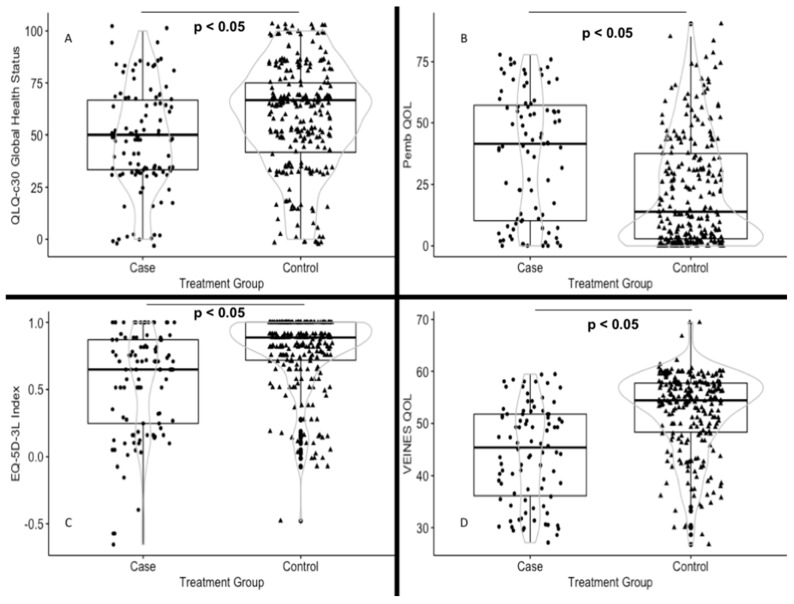
Cases and controls distribution of the data and its probability density in different questionnaires. Violin plots. (**A**) QLQ-c30 Global Health Status, (**B**) Pemb-QOL, (**C**) EQ-5D-3L Index, and (**D**) VEINES QOL. The plots show the actual distribution of the observed values (dots in cases and triangles in controls), ultimately allowing a representation of the scores used in the study. Boxplots depicts the median (black line inside the box) and the interquartile range (the edges of the box). The kernel density plot (grey line) represents the probability density function. Larger sections of the violin plot indicate a higher probability of observation at a given value, whereas thinner sections correspond to a lower probability.

**Table 1 cancers-12-00075-t001:** Clinical characteristics of the study patients.

Variable	Cases (Cancer with VTE)	Controls (Cancer without VTE)	Entire Cohort
Male sex, n (%)	65 (50.8%)	180 (60.8%)	245 (57.8%)
Age, years (n = 425); mean ± SD	63 (11.4)	59.1 (20.6)	60.2 (18.4)
Body mass index, kg/m^2^ (n = 321), mean ± SD	27.2 (5.7)	26.1 (4.9)	26.4 (5.2)
Arterial hypertension (n = 392), n (%)	55 (44.0%)	88 (33.0%)	143 (36.5%)
Dyslipidemia (n = 391), n (%)	33 (26.6%)	58 (21.7%)	91 (23.3%)
Diabetes mellitus (n = 392), n (%)	18 (14.4%)	43 (16.1%)	61 (15.6%)
Asthma (n = 392), n (%)	3 (2.4%)	4 (1.5%)	7 (1.8%)
Acute coronary syndrome (n = 392), n (%)	3 (2.4%)	6 (2.2%)	9 (2.3%)
Stroke (n = 392), n (%)	6 (4.8%)	7 (2.6%)	13 (3.3%)
Smoking (n = 392), n (%)	30 (24.0%)	61 (22.8%)	91 (23.2%)
Non-steroidal anti-inflammatory drugs (n = 392), n (%)	12 (9.6%)	20 (7.5%)	32 (8.2%)
Statins (n = 392), n (%)	22 (17.6%)	32 (12.0%)	54 (13.8%)
Active anticancer treatment, n (%)	95 (75.4%)	203 (77.5%)	298 (76.8%)
Central venous catheter, n (%)	7 (5.6%)	70 (26.7%)	77 (19.8%)
**ECOG performance status (n = 364)**			
0, n (%)	30 (25.4%)	105 (42.7%)	135 (37.1%)
1, n (%)	65 (55.1%)	125 (50.8%)	190 (52.2%)
2, n (%)	17 (14.4%)	13 (5.3%)	30 (8.2%)
3, n (%)	5 (4.2%)	3 (1.2%)	8 (2.2%)
4, n (%)	1 (0.8%)	0 (0.0%)	1 (0.3%)
Metastasis (n = 341), n (%)	80 (64.5%)	173 (69.8%)	253 (68%)
**Tumor site (n = 425)**			
Gynecologic, n (%)	24 (18.8%)	19 (6.4%)	43 (10.1%)
Lung, n (%)	37 (28.9%)	47 (15.8%)	84 (19.8%)
Digestive, n (%)	18 (14.1%)	82 (27.6%)	100 (23.5%)
Genitourinary, n (%)	11 (8.6%)	17 (5.7%)	28 (6.6%)
Lymphoma, n (%)	9 (7%)	49 (16.5%)	58 (13.6%)
Other sites, n (%)	29 (22.7%)	83 (27.9%)	112 (26.4%)

Abbreviations: VTE: venous thromboembolism; SD: standard deviation; HDL: high-density lipoprotein; LDL: low-density lipoprotein; ECOG: Eastern Cooperative Oncology Group.

**Table 2 cancers-12-00075-t002:** Differences in the Pemb-QOL, EORTC QLQ-C30 functional and symptoms scales, EQ-5D-3L, and VEINES scale scores between cases and controls.

**Pemb-QOL ^1^**	**Cases (Cancer with VTE)**	**Controls (Cancer without VTE)**	**Mean Difference**	**Effect Size (95% CI)**	**Relative Efficiency (95% CI)**
Quality of life	44.4	23.0	+21.4 **	0.99 (0.6; 1.5)	0.93 (0.6; 1.2)
Frequency of complaints	33.6	21.0	+12.6 **	0.86 (0.6; 1.4)	0.51 (0.2; 0.8)
Activities daily living limitations	52.1	29.3	+22.8 **	1.06 (0.7; 1.7)	0.79 (0.4; 1.1)
Work-related problems	72.9	39.5	+33.4 **	0.84 (0.5; 1.3)	0.74 (0.4; 1.0)
Social limitations	37.2	15.5	+21.7 **	1.47 (0.9; 2.3)	0.78 (0.4; 1.1)
Intensity of complaints	33.9	15.5	+18.4 **	1.32 (0.9; 2.1)	0.86 (0.5; 1.1)
Emotional complaints	36.4	17.0	+19.4 **	1.33 (0.9; 2.1)	0.89 (0.6; 1.2)
**EORTC QLQ-C30 Functional Scale ^2^**	**Cases (Cancer with VTE)**	**Controls (Cancer without VTE)**	**Mean Difference**	**Effect Size (95% CI)**	**Relative Efficiency (95% CI)**
Global health status	47.7	58.4	−10.3 **	0.95 (0.7; 1.3)	−0.42 (−0.6; −0.2)
Physical functioning	64.4	80.6	−16.2 **	1.29(0.9; 1.8)	−0.70 (−0.9; −0.4)
Role functioning	53.9	74.7	−20.8 **	1.25 (0.9; 1.7)	−0.65 (−0.9; −0.4)
Emotional functioning	67.2	72.3	−5.1	0.94 (0.7; 1.3)	−0.2 (−0.04; 0.1)
Cognitive functioning	81.7	84.4	−2.7	1.37 * (1.0; 1.9)	−0.11 (−0.3; 0.1)
Social functioning	62.2	74.1	−11.9 **	1.35 (0.9; 1.8)	−0.39 (−0.6; −0.1)
**EORTC QLQ-C30 symptoms scale ^3^**	**Cases (Cancer with VTE)**	**Controls (Cancer without VTE)**	**Mean Difference**	**Effect Size (95% CI)**	**Relative Efficiency (95% CI)**
Fatigue	48.4	33.7	+14.7 **	1.27 (0.9; 1.7)	0.52 (0.3; 0.7)
Nausea and vomiting	14.2	11.5	+2.7	0.98 (0.7; 1.3)	0.12 (−0.1;0.3)
Pain	36.0	23.0	+13.0 **	1.48 * (1.0; 2.0)	0.4 (0.2; 0.7)
Dyspnea	23.5	13.0	+10.5 **	1.5 ** (1.1; 2.0)	0.39 (0.2; 0.6)
Insomnia	30.1	31.9	−1.8	0.97 (0.7; 1.3)	−0.05 (−0.2;0.2)
Appetite loss	25.8	25.3	+0.5	1.0 (0.7; 1.4)	0.01 (−0.2;0.2)
Constipation	26.7	22.1	+4.6	1.25 (0.9; 1.7)	0.1 (−0.1;0.4)
Diarrhea	15.1	16.5	+1.4	1.21 (0.9; 1.7)	−0.05 (−0.3;0.2)
Financial difficulties	23.4	17.8	+5.6	1.35 (0.9; 1.9)	0.2 (−0.03;0.4)
**EQ-5D-3L ^4^**	**Cases** **(Cancer with VTE)**	**Controls** **(Cancer without VTE)**	**Mean Difference**	**Effect Size** **(95% CI)**	**Relative Efficiency** **(95% CI)**
Index score	0.55	0.77	−0.22 **	2.06 ** (1.5; 2.8)	−0.7 (−0.9; −0.5)
**VEINES-QOL/Sym ^5^**	**Cases** **(Cancer with VTE)**	**Controls** **(Cancer without VTE)**	**Mean Difference**	**Effect Size** **(95% CI)**	**Relative Efficiency** **(95% CI)**
VEINES-QOL	42.7	51.7	−9.0 **	0.82 (0.53; 1.20)	1.0 (0.79; 1.38)
VEINES/Sym	43.8	51.4	−7.6 **	0.80 (0.5–1.2)	0.8 (0.5–1.0)

Abbreviations: VTE: venous thromboembolism; CI: confidence interval. * *p* < 0.05; ** *p* < 0.001. ^1^ Pemb-QOL: Higher scores reflect a lower quality of life and the minimally clinically important difference is 15 points; ^2^ EORTC QLQ-C30 functional scale: Higher scores reflect a better health status and the minimally important difference varies from 6 to 15 points; ^3^ EORTC QLQ-C30 symptoms scale: Higher scores in the symptoms scale indicate a reduction in quality of life; ^4^ EQ-5D-3L: Higher scores reflect a better health status and the minimally important difference on the EQ-5D questionnaire for patients with malignancies is 0.06−0.08 UI; ^5^ VEINES-QOL/Sym: Higher scores reflect a better health status.

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
