# Peer review of "A Case-Control Analysis of the Impact of Venous Thromboembolic Disease on Quality of Life of Patients with Cancer: Quality of Life in Cancer (Qca) Study"

_cancers, 2019, doi:10.3390/cancers12010075_

Round 1

Reviewer 1 Report

Lucia Marin-Barrera and colleagues present a prospective cohort study, evaluating the quality of life in cancer patients with and without VTE. The topic is highly important as the consequences of VTE on the quality of life are not yet fully elucidated. The impact of VTE on quality of life is important for policy makers, clinicians and patients. 

One of the reasons that the subject is underrepresented in current literature, is that studying the impact of VTE on quality of life in cancer patients comes with methodological issues. My major concern for this study is that these methodological issues are not fully addressed by current study.

Major comments:

It is currently not clear during what time-point around the VTE event patients filled in the questionnaires. A patient will probably report worse QoL in the first couple of days after DVT/PE than after 3 weeks. Readers cannot appreciate the time to measurement which could bias the results. It is currently not clear at what time point after cancer diagnosis cases and controls are included in the study. This is highly important as there will be major differences in QoL between cancer patients. The authors did not report on time from cancer diagnosis or whether patients received curative treatment, (palliative) chemotherapy, surgery, etc. These factors presumably affect quality of life  more than PE/DVT, and thus should be adjusted for in the analyses.   In line with the previous comments, I am worried about the design which did not match cases and controls.Consequently, important differences between the two study group are present, which could explain some of the differences in QoL: there are substantial differences for age, lung cancer, and gynecologic cancer between the groups. It is unclear why these specific control patients were selected.

Some additional/minor comments:

The authors state that VTE is the second leading cause of death in cancer patients. In the accompanying reference, VTE is not the second leading cause. Please rephrase or use a different convincing reference. Please rephrase the second sentence of the methods as it is unclear: 'the study procedures involved..... In the result section it is stated: 'all of the study patients were receiving weight-adjusted LMWH at inclusion. I assume this only applies for the cases. Please adjust. The baseline table comprises many rudimentary variables: weigh/height (bmi is reported), blood pressure, and several blood tests which are not further discussed in the manuscript. I suggest to remove them from the table.  In table two, there is a blank space between the words 'cancer' and 'with VTE' in the column title. It seems that QoL is higher in those with VTE according to the Pemb-QOL questionnaire? 44.4 vs 23.3 Please comment on this observation in the discussion.  Remarkably, more work-related problems are reported in the cases (+33). Please comment on this, could it affect the results? I am not sure what figure 1 adds to the manuscript. Also, the left lower tile shows a file specification of the figure.    Throughout the file, the authors state that they objectified the association between VTE and Qol, or even the impact of VTE on QoL. This seriously needs to be toned down given the current study design which does not provide room to say anything about impact.  I currently miss the post-hoc analysis of the Catch trial which evaluated QoL in the discussion section. Please consider adding it. Pubmed ID: 29680102  

Author Response

Ms. Ref. No.: cancers-673688

Title: A case-control analysis of the impact of venous thromboembolic disease on quality of life of patients with cancer: Quality of life in Cancer (Qca) study

Juliana Zhang

Assistant Editor

Cancers

Dear Dr. Zhang,

I thank you for the opportunity to revise and re-submit our manuscript cancers-673688. We also grateful to the Reviewers for their insightful comments, suggestions, and questions, which we believe have helped to significantly strengthen our article. Several valid points were raised in the review and, after careful consideration, we have made the requisite revisions to the manuscript. Below, you will find our point-by-point responses to the Reviewers’ comments, questions, and concerns; the specific changes to the manuscript are highlighted. All authors have agreed to the changes in the revised paper.

We sincerely hope that our responses and revisions have adequately addressed the Reviewers’ concerns, and that our manuscript is now suitable for publication in Cancers.

Sincerely,

Luis Jara-Palomares, MD, PhD

AUTHORS’ RESPONSE TO REVIEWER’S #1 COMMENTS

Q1. Moderate English changes required 

REPLY. This work was edited and reviewed by an expert person in the writing of medical texts in English. Even so, after the work done to prepare the responses to the reviewers, the text has been reviewed again.

Q2. It is currently not clear during what time-point around the VTE event patients filled in the questionnaires. A patient will probably report worse QoL in the first couple of days after DVT/PE than after 3 weeks. Readers cannot appreciate the time to measurement which could bias the results. It is currently not clear at what time point after cancer diagnosis cases and controls are included in the study. This is highly important as there will be major differences in QoL between cancer patients.

REPLY We appreciate the Reviewer’s highly relevant remark. In the design of the study, the analysis of the quality of life in patients with cancer and thrombosis (cases) was performed 30 days after the thrombotic event (30 +/- 2 days). We have clarified this point in the methods section, page 3, first paragraph: “Cases were included when they presented with acute symptomatic VTE, and the quality of life questionnaires were filled in 30 days after the thrombotic event (30 +/- 2 days), and all questionnaires were completed at the same time to be able to compare the results of quality of life of each questionnaire at the same vital moment of the patient.”

Q3. The authors did not report on time from cancer diagnosis or whether patients received curative treatment, (palliative) chemotherapy, surgery, etc. These factors presumably affect quality of life more than PE/DVT, and thus should be adjusted for in the analyses. In line with the previous comments, I am worried about the design which did not match cases and controls. Consequently, important differences between the two-study group are present, which could explain some of the differences in QoL: there are substantial differences for age, lung cancer, and gynecologic cancer between the groups. It is unclear why these specific control patients were selected.

REPLY. We thank the Reviewer for raising this cogent point. Indeed, due to space problems we did not include any of the data that the reviewer requires. There were no differences between the time since the diagnosis of cancer and the inclusion of cases and controls in the study, with a mean of 14.76 +/- 21.34 months and 15.01 +/- 25.57 months, respectively. More than half of the cases (54.5%) and controls (58.1%) were included in the first 6 months after the cancer diagnosis.

In previously published works, analysis of quality of life with a specific disease was performed and subsequently these results were compared with a historical cohort. This type of design is simpler, but it has a major limitation, since populations can be different from an epidemiological, social point of view and even with different options of therapy. The approach of our study (case-controls) lay in the need to compare in similar populations, and in the same period of time the quality of life in order to obtain more robust results. The main objective of the study (primary objective) was to analyze the impact of thrombosis on the quality of life and quality of cancer patients. Based on this objective, the sample size was raised. With the proposed design, we have managed to obtain similar populations, although given that perfect case-control matching cannot be performed over time, there are variables that were not perfectly adjusted. Obviously, for logistical reasons, it is impossible to carry out a matched for each and every one of the characteristics of the population.

We have clarified this point:

In the “results” section as follows: “There were no differences between the time since the diagnosis of cancer and the inclusion of cases and controls in the study, with a mean of 14.76 +/- 21.34 months and 15.01 +/- 25.57 months, respectively. More than half of the cases (54.5%) and controls (58.1%) were included in the first 6 months after the cancer diagnosis.”

In the “Discussion” section as follows: “Although the design of our work was a case-control study, the prospective and multicentre development has not allowed us to be able to adjust all the variables and that could have affected the score in the quality of life (for example, the locations of the neoplasms it is not fully balanced in cases and controls). Even so, in all cancer locations the presence of thrombosis had a negative impact. Even so, more than half of the cases (54.5%) and controls (58.1%) were included in the first 6 months after the cancer diagnosis, and in both groups the percentage of metastases was similar, so the treatment options in both groups should not have differed much”.

Q4. The authors state that VTE is the second leading cause of death in cancer patients. In the accompanying reference, VTE is not the second leading cause. Please rephrase or use a different convincing reference.

REPLY. We thank the Reviewer for the pertinent comment. We apologize for the reference used in this sentence and we have modified that reference.

Noble S, Pasi J. Epidemiology and pathophysiology of cancer-associated thrombosis. Br J Cancer. 2010;102(Suppl 1):S2-9.

Q5. Please rephrase the second sentence of the methods as it is unclear: 'the study procedures involved.....

REPLY. We thank the Reviewer for the pertinent suggestion. We have rephrased the sentence: “Investigators were specialists in oncology, hematology, internal medicine, and respiratory medicine”.

Q6. In the result section, it is stated: 'all of the study patients were receiving weight-adjusted LMWH at inclusion. I assume this only applies for the cases. Please adjust.

REPLY. We apologize for the inconvenience. It is true that only cases received LMWH. For that reason, the following text was added: “All cases received weight-adjusted low molecular weight heparin at inclusion”

Q7. The baseline table comprises many rudimentary variables: weigh/height (bmi is reported), blood pressure, and several blood tests which are not further discussed in the manuscript. I suggest to remove them from the table. 

REPLY. We thank the Reviewer for this comment. For suggestion, we have removed some variables in table 1.

Q8. In table two, there is a blank space between the words 'cancer' and 'with VTE' in the column title.

It seems that QoL is higher in those with VTE according to the Pemb-QOL questionnaire? 44.4 vs 23.3 Please comment on this observation in the discussion.  Remarkably, more work-related problems are reported in the cases (+33). Please comment on this, could it affect the results?

REPLY. We apologize for the format of the table, but there is no blank space where the reviewer suggests. We have tried to modify the format so that the table can be read properly.

I appreciate the reviewer comment on the Pemb-QOL analysis. It is very important to highlight that the interpretation of QoL questionnaires are not always in the same way. In the P-Emb-QoL questionnaire, a higher score implies a worse quality of life. This aspect had been previously commented on in the methodology (page 3, last paragraph and page 4, first paragraph): “Higher scores reflect a lower QoL and the minimally clinically important difference (MCID) is 15 points”

Even so, to clarify this point and avoid confusions for readers, we have included at the bottom of the table 2 a summary to make easier adequate interpretation of the results.

Table 7, Table footer, page 7:

1 Pemb-QOL: Higher scores reflect a lower quality of life and the minimally clinically important difference is 15 points; 2 EORTC QLQ-C30 functional scale: Higher scores reflect a better health status and minimally important difference varies from 6 to 15 points; 3 EORTC QLQ-C30 symptoms scale: Higher scores in the symptoms scale indicate a reduction in quality of life; 4 EQ-5D-3L: Higher scores reflect a better health status and the minimally important difference on the EQ-5D questionnaire for patients with malignancies is 0.06−0.08 UI; 5 VEINES-QOL/Sym: Higher scores reflect a better health status.”

Q9. I am not sure what figure 1 adds to the manuscript. Also, the left lower tile shows a file specification of the figure.  Throughout the file, the authors state that they objectified the association between VTE and Qol, or even the impact of VTE on QoL. This seriously needs to be toned down given the current study design which does not provide room to say anything about impact. 

REPLY. This is a highly relevant point. Figure 1 is essential in this work. It is true that with the format of this journal appear a file specification of the figure. This problem has been solved.

Violin plots is a method of plotting numeric data and can be considered a combination of the box plot with a kernel density plot. Kernel density estimation (KDE) is a non-parametric way to estimate the probability density function of a random variable. Moreover, in this plots are showing the real distribution (dots in cases and triangle in controls) and with this representation we make an abstraction about the distribution of scores used in the study. The boxplots, shows us the median (black line inside the box) and the interquartile range (the box). The kernel density plot (grey line) represent the probability density function, wider sections of the violin plot represent a higher probability of observations taking a given value, the thinner sections correspond to a lower probability.

To clarify this point in “Methods/Statistical analysis” was included: “Figure was represented by violin plot. Violin plots is a method of plotting numeric data and can be considered a combination of the box plot with a kernel density plot. Kernel density estimation (KDE) is a non-parametric way to estimate the probability density function of a random variable. Moreover, in this plots are showing the real distribution (dots in cases and triangle in controls) and with this representation we make an abstraction about the distribution of scores used in the study. The boxplots, shows us the median (black line inside the box) and the interquartile range (the box). The kernel density plot (grey line) represent the probability density function, wider sections of the violin plot represent a higher probability of observations taking a given value, the thinner sections correspond to a lower probability.”

We have included in “results” section, page 7, legend figure 1:

“Plots are showing the real distribution (dots in cases and triangle in controls) and with this representation we make an abstraction about the distribution of scores used in the study. The boxplots, shows us the median (black line inside the box) and the interquartile range (the box). The kernel density plot (grey line) represent the probability density function, wider sections of the violin plot represent a higher probability of observations taking a given value, the thinner sections correspond to a lower probability.”

In “results” section, subheading “Impact of venous thromboembolism on quality of life of patients with malignancies”, page 5, we have added: “The presence of thrombosis in oncological patients had a statistically significant and clinically relevant negative impact on the four quality of life questionnaires analysed.”

We have included in “results” section, page 7, paragraph: “Figure 1 shows the four sections of the violin plot with differences in median between cases and controls (p < 0.05) and we can observe evident differences in the kernel distribution being the controls in the four questionnaires surrounding the median value and wider distribution in the cases.”

Q10. I currently miss the post-hoc analysis of the Catch trial which evaluated QoL in the discussion section. Please consider adding it. Pubmed ID: 29680102   

REPLY. We thank the Reviewer for the pertinent suggestion. In order to address the Reviewer’s concern, the following text was added to the “Discussion” section, page 9: “In 2018, a sub-analysis from CATCH trial was published [32]. This trial explored the role of the LMWH tinzaparin in preventing rVTE in patients with different types of cancer. In this study, the administration of EQ-5D was included, and this questionnaire was evaluated throughout the study. Authors found that recurrence VTE had a significant impact on EQ-5D scores (decrement: -0.075). The strengths of this study are the sample size (n = 883) and the repeated measurement of the quality of life in patients (every month, for 7 months). The work presents the limitation that it does not analyze specific quality of life questionnaires, and that the comparison of quality of life was based on a model with a specific patient profile (male, from Western Europe, no distant metastases, symptomatic DVT as the qualifying event, and ECOG performance status of 1 at baseline).”

[32] Lloyd AJ, Dewilde S, Noble S, Reimer E, Lee AYY. What Impact Does Venous Thromboembolism and Bleeding Have on Cancer Patients' Quality of Life? Value Health. 2018;21(4):449-455.

Reviewer 2 Report

Very interesting research on quality of life, with a strong design and broad data collection.

Overall remark: I miss some clinical interpretation of the results and a comment on how to improve the care/support for these patients based on the findings of this paper. It would be good to complete the results of the quantitative study, with some qualitative data. Can you add some clinical impact, and comment on the prospects of additional research in the discussion?

Methods:

Questionnaires and statistics are very well described.

Please add a section on the ethics in the study: approval of ethics committee, informed consent, burden for patients to complete the questionnaires. Did all patients complete al questionnaires?

Please also add a section on how patients were recruited. What was the respons in this study on the different questionnaires?

When did the patients complete the questionnaires? How many times? All at once? Do you have any information on when in the disease trajectory patients were included in the study? Can this have influenced the results?

How were the patient data retrieved? 

Do you have any information on the differences between the centers that participated in the study? Could this potentially have infleunced the results? Did the respons differ per center? How many patients participated per center? Maybe add this information to the results section.

Results:

As mentioned above, please add some information on overall respons and respons per participating center. 

Table 2: did you control for criteria such as age, sex and localization of the malignancy? Or other patient/center characteristics?

Discussion: 

What is a clinical significant difference in your study? What can be done for the patients? At the end of this section you speak about a more holistic clinical management. What does that mean? Which are elements of this management?

Author Response

Ms. Ref. No.: cancers-673688

Title: A case-control analysis of the impact of venous thromboembolic disease on quality of life of patients with cancer: Quality of life in Cancer (Qca) study

Juliana Zhang

Assistant Editor

Cancers

Dear Dr. Zhang,

I thank you for the opportunity to revise and re-submit our manuscript cancers-673688. We also grateful to the Reviewers for their insightful comments, suggestions, and questions, which we believe have helped to significantly strengthen our article. Several valid points were raised in the review and, after careful consideration, we have made the requisite revisions to the manuscript. Below, you will find our point-by-point responses to the Reviewers’ comments, questions, and concerns; the specific changes to the manuscript are highlighted. All authors have agreed to the changes in the revised paper.

We sincerely hope that our responses and revisions have adequately addressed the Reviewers’ concerns, and that our manuscript is now suitable for publication in Cancers.

Sincerely,

Luis Jara-Palomares, MD, PhD

AUTHORS’ RESPONSE TO REVIEWER’S #2 COMMENTS

Q1. Very interesting research on quality of life, with a strong design and broad data collection.

Overall remark: I miss some clinical interpretation of the results and a comment on how to improve the care/support for these patients based on the findings of this paper. It would be good to complete the results of the quantitative study, with some qualitative data. Can you add some clinical impact, and comment on the prospects of additional research in the discussion?

REPLY. We thank the Reviewer for raising this appropriate concern.

We have included in “discussion” section, first paragraph: “The practical interpretation of our study is that, due to there is a negative impact on the quality of life in oncological patients with thrombosis, more emphasis should be placed on research and education in health personnel to identify those aspects that affect the quality of life. In addition, identifying clinically relevant differences can help us assess the clinical effectiveness of treatments and interventions performed in patients with cancer associated thrombosis.”

For that reason, we have included in “discussion” section, fourth paragraph:

“The data obtained with the quality of life questionnaires provide us interesting and reproducible information in different dimensions of the patient. Likewise, the information obtained with this type of studies can be complemented with qualitative research work. In this sense, it is worth emphasizing the work published by Seaman et al. in which they conducted semi-structured interviews in 14 patients with CAT [33]. In this study, patients considered that VTE had a major impact on their life, considering as a distinct entity rather more than a part of their cancer illness. They identified three areas in which VTE affected their lives: 1) symptom burden of VTE; 2) The impact in the context of their cancer journey; 3) Impact on their activities of daily living.”

[33] Noble S, Prout H, Nelson A. Patients' Experiences of LIving with CANcer-associated thrombosis: the PELICAN study. Patient Prefer Adherence. 2015;9:337-45.

Q2. Methods: Questionnaires and statistics are very well described. Please add a section on the ethics in the study: approval of ethics committee, informed consent, burden for patients to complete the questionnaires. Did all patients complete al questionnaires? Please also add a section on how patients were recruited. What was the response in this study on the different questionnaires? When did the patients complete the questionnaires? How many times? All at once?

REPLY. We thank the Reviewer for raising these issues. Due to all these questions are related, we have answered all of them in this point

Information regarding informed consent and how patients were recruited is found in the "Methods / study design" section, first paragraph

Regarding the response of the patients to the questionnaires, we have included this information in the section “Results Impact of venous thromboembolism on quality of life of patients with malignancies”

Of the 425 patients included, 84% completed the EQ-5D questionnaire (n=355), 83% completed the EORTC QLQ-C30 questionnaire (n=354), 72% completed the Pemb-QOL questionnaire (n=306), and 72% completed the VEINES-QOL / Sym questionnaire (n=305).

The following text was added to the “Methods/Study design” section, first paragraph

“Cases were included when they presented with acute symptomatic VTE, and the quality of life questionnaires were filled in 30 days after the thrombotic event (30 +/- 2 days), and all questionnaires were completed at the same time to be able to compare the results of quality of life of each questionnaire at the same vital moment of the patient.”

Q3. Do you have any information on when in the disease trajectory patients were included in the study? Can this have influenced the results?

REPLY. We thank the Reviewer for raising this point. Indeed, due to space problems we did not include any of the data that the reviewer requires. There were no differences between the time since the diagnosis of cancer and the inclusion of cases and controls in the study, with a mean of 14.76 +/- 21.34 months and 15.01 +/- 25.57 months, respectively. More than half of the cases (54.5%) and controls (58.1%) were included in the first 6 months after the cancer diagnosis.

We have clarified this point:

In the “results” section as follows: “There were no differences between the time since the diagnosis of cancer and the inclusion of cases and controls in the study, with a mean of 14.76 +/- 21.34 months and 15.01 +/- 25.57 months, respectively. More than half of the cases (54.5%) and controls (58.1%) were included in the first 6 months after the cancer diagnosis.”

Q4. How were the patient data retrieved? 

REPLY. We appreciate the Reviewer’s comment. The following text was added to the “Methods/Study design” section, first paragraph: “All clinical data and questionnaires were included by the researchers in an anonymized database.”

Q5. Do you have any information on the differences between the centers that participated in the study? Could this potentially have influenced the results? Did the respons differ per center? How many patients participated per center? Maybe add this information to the results section. Results: As mentioned above, please add some information on overall respons and respons per participating center. 

REPLY. We appreciate the Reviewer’s comment. There were differences between the centers because some centers recruited only cases, others only controls and others cases and controls. We have included a supplemental table 9 to provide this information. The distribution of the centers was representative from different parts of Spain. Differences responses between center could be possible, but it was not the objective of our study and we need to avoid overmatching because it could imply an erroneous interpretation of the results.

Q6. Table 2: did you control for criteria such as age, sex and localization of the malignancy? Or other patient/center characteristics?

REPLY. We appreciate the Reviewer’s highly relevant remark.

In previously published works, analysis of quality of life with a specific disease was performed and subsequently these results were compared with a historical cohort. This type of design is simpler, but it has a major limitation, since populations can be different from an epidemiological, social point of view and even with different options of therapy. The approach of our study (case-controls) lay in the need to compare in similar populations, and in the same period of time the quality of life in order to obtain more robust results. The main objective of the study (primary objective) was to analyze the impact of thrombosis on the quality of life and quality of cancer patients. Sample size was calculated to find differences in QoL between groups. With the proposed design, we have tried to obtain similar populations over the time. Subgroup analyses were done (PE vs. DVT; male vs. female; cancer location) and described in results section, although this analyses were secondary objectives. We did not plan to do a controlled analysis according to different criteria (age, sex, metastases, localization, ECOG, oncological treatment etc.) because there are a lot of possibilities on this way and sample size was not enough to do this. We talk this possibility with our statistician and this kind of analyses implies a risk of overmatching that could imply an erroneous interpretation of the results.

We have included in the “Discussion” section as follows: “Although the design of our work was a case-control study, the prospective and multicentre development has not allowed us to be able to control by all variables that could have affected the score in the quality of life (for example, the locations of the neoplasms it is not fully balanced in cases and controls). The performance of an analysis controlling for different variables (age, sex, metastases, localization, ECOG, oncological treatment etc.) was not possible because the sample size was not calculated in order to perform these analyses. The execution of analyses controlled by several variables implies a risk of overmatching that could imply an erroneous interpretation of the results. Even so, in all cancer locations the presence of thrombosis had a negative impact. In addition, more than half of the cases (54.5%) and controls (58.1%) were included in the first 6 months after the cancer diagnosis, and in both groups the percentage of metastases was similar, so the treatment options in both groups should not have differed much.”.

Q7. Discussion: What is a clinical significant difference in your study?

REPLY. We thank the Reviewer for the pertinent comment. In our work we evaluated four quality of life questionnaires. As indicated in the "Methods" section, the minimally important difference (MID) on the EQ-5D questionnaire for patients with malignancies (0.06−0.08 UI) [19]. The MID on the EORTC QLQ-C30 questionnaire varies from 6 to 15 points [23-25]. The minimally clinically important difference (MCID) in the PEmb-QoL questionnaire is 15 points [26]. We have based the criteria of clinically significant differences on those criteria previously published.

Q8. What can be done for the patients? At the end of this section you speak about a more holistic clinical management. What does that mean? Which are elements of this management?

REPLY. We thank the Reviewer for raising this appropriate concern. As we have commented above the practical interpretation of our study is that, due to there is a negative impact on the quality of life in oncological patients with thrombosis, more emphasis should be placed on research and education in health personnel to identify those aspects that affect the quality of life. In addition, identifying clinically relevant differences can help us assess the clinical effectiveness of treatments and interventions performed in patients with cancer associated thrombosis.

We have included in “discussion” section, first paragraph: “The practical interpretation of our study is that, due to there is a negative impact on the quality of life in oncological patients with thrombosis, more emphasis should be placed on research and education in health personnel to identify those aspects that affect the quality of life. In addition, identifying clinically relevant differences can help us assess the clinical effectiveness of treatments and interventions performed in patients with cancer associated thrombosis.”

With “holistic” expression we wanted to say that a comprehensive and complete vision must be used in the analysis of the disease. Clinicians who treat cancer associated thrombosis should not only focus on the patient not having a hemorrhage or a recurrence, but we should have a more comprehensive view of the patient, since the patient's situation has a negative impact on their quality of life and generates anguish.

To clarify this point, we have modified this conclusion as follows: “Clinicians who treat cancer associated thrombosis should have a comprehensive view of the patient. The quality of life questionnaires will allow us to evaluate the impact of those act destined to improve the quality of life of patients with cancer associated thrombosis.

Reviewer3

Q9. I am not sure what figure 1 adds to the manuscript. Also, the left lower tile shows a file specification of the figure.  Throughout the file, the authors state that they objectified the association between VTE and Qol, or even the impact of VTE on QoL. This seriously needs to be toned down given the current study design which does not provide room to say anything about impact. 

REPLY. This is a highly relevant point. Figure 1 is essential in this work. It is true that with the format of this journal appear a file specification of the figure. This problem has been solved.

Violin plots is a method of plotting numeric data and can be considered a combination of the box plot with a kernel density plot. Kernel density estimation (KDE) is a non-parametric way to estimate the probability density function of a random variable. Moreover, in this plots are showing the real distribution (dots in cases and triangle in controls) and with this representation we make an abstraction about the distribution of scores used in the study. The boxplots, shows us the median (black line inside the box) and the interquartile range (the box). The kernel density plot (grey line) represent the probability density function, wider sections of the violin plot represent a higher probability of observations taking a given value, the thinner sections correspond to a lower probability.

To clarify this point in “Methods/Statistical analysis” was included: “Figure was represented by violin plot. Violin plots is a method of plotting numeric data and can be considered a combination of the box plot with a kernel density plot. Kernel density estimation (KDE) is a non-parametric way to estimate the probability density function of a random variable. Moreover, in this plots are showing the real distribution (dots in cases and triangle in controls) and with this representation we make an abstraction about the distribution of scores used in the study. The boxplots, shows us the median (black line inside the box) and the interquartile range (the box). The kernel density plot (grey line) represent the probability density function, wider sections of the violin plot represent a higher probability of observations taking a given value, the thinner sections correspond to a lower probability.”

We have included in “results” section, page 7, legend figure 1:

“Plots are showing the real distribution (dots in cases and triangle in controls) and with this representation we make an abstraction about the distribution of scores used in the study. The boxplots, shows us the median (black line inside the box) and the interquartile range (the box). The kernel density plot (grey line) represent the probability density function, wider sections of the violin plot represent a higher probability of observations taking a given value, the thinner sections correspond to a lower probability.”

In “results” section, subheading “Impact of venous thromboembolism on quality of life of patients with malignancies”, page 5, we have added: “The presence of thrombosis in oncological patients had a statistically significant and clinically relevant negative impact on the four quality of life questionnaires analysed.”

We have included in “results” section, page 7, paragraph: “Figure 1 shows the four sections of the violin plot with differences in median between cases and controls (p < 0.05) and we can observe evident differences in the kernel distribution being the controls in the four questionnaires surrounding the median value and wider distribution in the cases.”

Round 2

Reviewer 1 Report

Dear author, 

In my opinion the manuscript has improved due to the adjustments made by authors. My major concern however remains:

The study assumes that VTE explains the difference in QoL between both groups. However, many other variables in cancer patients can affect QoL. Many of these variables have not been described by authors: chemotherapy use, episodes of neutropenic fever, infections, recent hospitalizations. Also it is not described whether patients received curative or palliative treatment, or what the 12-month survival probability was in both groups. Please either describe these factors, or diclose their absence in the limitation section. 

One other minor point: the suggested reference of the authors also does not contain evidence that VTE is the second cause of death in cancer patients. Please tone down the statement made in the manuscript. 

Author Response

Ms. Ref. No.: cancers-673688

Title: A case-control analysis of the impact of venous thromboembolic disease on quality of life of patients with cancer: Quality of life in Cancer (Qca) study

Juliana Zhang

Assistant Editor

Cancers

Dear Dr. Zhang,

Thank you very much for your email. On the next paragraphs, we are addressing the Reviewer´s comments and concerns.

We sincerely hope that the current version might be acceptable for publication in Cancers your Journal.

Sincerely,

Luis Jara-Palomares, MD, PhD

AUTHORS’ RESPONSE TO REVIEWER’S COMMENTS

Q1. In my opinion the manuscript has improved due to the adjustments made by authors. My major concern however remains: The study assumes that VTE explains the difference in QoL between both groups. However, many other variables in cancer patients can affect QoL. Many of these variables have not been described by authors: chemotherapy use, episodes of neutropenic fever, infections, recent hospitalizations. Also, it is not described whether patients received curative or palliative treatment, or what the 12-month survival probability was in both groups. Please either describe these factors, or disclose their absence in the limitation section.

REPLY. We thank the Reviewer for raising this cogent point. This point has been clarified in the revised “Discussion” section, as follows: “Moreover, there were other variables that could affect to QoL that were not included in our study (i. e. chemotherapy use, episodes of neutropenic fever, infections, recent hospitalizations, if patients received curative or palliative treatment or 12-month survival probability in both groups)”

Q2. One other minor point: the suggested reference of the authors also does not contain evidence that VTE is the second cause of death in cancer patients. Please tone down the statement made in the manuscript. 

REPLY. We appreciate the Reviewer’s comment. In order to address the Reviewer’s comment, the sentence has been changed as follows: “The burden of this condition is even higher in patients with malignancies”, instead of: “The burden of this condition is even higher in patients with malignancies – in whom it is the second most common cause of death after cancer progression”
